# Bacteria as Nanoparticle Carriers for Immunotherapy in Oncology

**DOI:** 10.3390/pharmaceutics14040784

**Published:** 2022-04-03

**Authors:** Víctor M. Moreno, Alejandro Baeza

**Affiliations:** 1Departamento de Química en Ciencias Farmacéuticas, Facultad de Farmacia, Universidad Complutense de Madrid, Instituto de Investigación Sanitaria, Hospital 12 de Octubre i+12, Plaza Ramón y Cajal s/n, 28040 Madrid, Spain; victomor@ucm.es; 2Departamento Materiales y Producción Aeroespacial, ETSI Aeronáutica y del Espacio, Universidad Politécnica de Madrid, 28040 Madrid, Spain

**Keywords:** immunotherapy, bacterial carriers, nanomedicine, oncology

## Abstract

The use of nanocarriers to deliver antitumor agents to solid tumors must overcome biological barriers in order to provide effective clinical responses. Once within the tumor, a nanocarrier should navigate into a dense extracellular matrix, overcoming intratumoral pressure to push it out of the diseased tissue. In recent years, a paradigm change has been proposed, shifting the target of nanomedicine from the tumoral cells to the immune system, in order to exploit the natural ability of this system to capture and interact with nanometric moieties. Thus, nanocarriers have been engineered to interact with immune cells, with the aim of triggering specific antitumor responses. The use of bacteria as nanoparticle carriers has been proposed as a valuable strategy to improve both the accumulation of nanomedicines in solid tumors and their penetration into the malignancy. These microorganisms are capable of propelling themselves into biological environments and navigating through the tumor, guided by the presence of specific molecules secreted by the diseased tissue. These capacities, in addition to the natural immunogenic nature of bacteria, can be exploited to design more effective immunotherapies that yield potent synergistic effects to induce efficient and selective immune responses that lead to the complete eradication of the tumor.

## 1. Introduction

The use of nanoparticles as carriers of chemotherapeutic agents in oncology has received huge attention in recent decades [1]. The cornerstone of this interest is based on the discovery of their tendency to accumulate in solid tumors—the well-known enhanced permeation and retention (EPR) effect [2]. Solid tumors require large amounts of nutrients and, therefore, they force the rapid fabrication of blood vessels in the diseased tissue [3]. These vessels present large pores of a few hundreds of nanometers, through which the nanocarriers can penetrate into the tissue, whereas they are not able to leak out from healthy blood vessels. This elegant conception has boosted the development of a myriad of nanomaterials, most of them endowed with elaborate stimulus-responsive properties, to deliver cytotoxic drugs to solid tumors in a controlled and selective manner [4]. Despite the huge effort carried out by the scientific community to produce smarter and safer nanomedicines, only a few of them have reached the market [5]. A nanocarrier should overcome several biological barriers in order to be effective [6]. First, it must avoid capture by the cells of the macrophagocytic system (MPS), which are trained to recognize and capture foreign moieties. Nanoparticle surface decoration with hydrophilic polymers such as polyethylene glycol (PEG) [7], zwitterionic shells [8], or even cell membrane coatings [9] has been widely employed to provide stealth properties against immune cells. Second, the nanocarriers should arrive at the solid tumors while avoiding their off-target accumulation in other organs. The conventional dogma states that tumor accumulation is driven by the EPR effect, but in recent years, this idea has received intense criticism [10]. The EPR effect is intense in murine models, but not so abundant in human tumors [11]. In any case, there are pharmacological and physical interventions that enhance this effect, improving the accumulation of the nanoparticles in solid tumors [12]. Once the nanocarriers reach the tumor, they should navigate through the tissue to distribute homogeneously throughout the diseased tissue. The diffusion rate of a nanometric entity in tumoral tissue is extremely low, especially considering the dense extracellular matrix and the existence of high intratumoral pressure, which push the nanoparticles away [13]. Different strategies have been proposed to overcome the poor penetration, from the use of proteolytic enzymes to digest the extracellular matrix [14], to the use of ultrasound to propel the nanocarriers within the tissue [15]. Finally, the nanoparticles must be engulfed by the tumoral cells. This is not a simple task, because solid tumors are composed of many different cell populations—not just tumoral ones. The conventional approach is to attach on the nanocarrier surface molecules or macromolecules that selectively bind membrane receptors overexpressed by tumoral cells. This strategy is known as active targeting [16]. However, the presence of these targeting moieties on the nanocarriers’ surface compromises their penetration into the tissue, because the nanocarriers are retained in the first tumoral cell layers due to the strong binding with their receptors—an effect known as binding site barrier [6]. An efficient nanocarrier should be designed to overcome all of these barriers. The complexity of this daunting task could explain the current poor outcomes of nanomedicines on the market. In the recent years, nano-oncology has experienced a paradigm shift from approaches centered on finding and destroying tumoral cells to strategies focused on the stimulation of the immune system of the patient in order to trigger a sustained response against the tumor [17]. This approach exploits the excellent capacities of immune cells to capture nanoparticles for “working with the biology” (taking these barriers into account and using them as advantages) instead of “working against the biology” (trying to overcome them) [18]. Thus, nanoparticles can be engineered to interact with immune cells instead of avoiding them. 

In this review, the basic principles of the interactions between tumoral cells and immune cells, as well as how they can be manipulated with nanoparticles, are presented. The use of bacteria as nanoparticle carriers has received increasing attention in recent years [19]. The self-motile and chemotactic capacity of bacteria has been exploited to deliver nanoparticles to deep zones of solid tumors [20]. Thus, the external surface of the bacterial wall has been functionalized with functional groups that allow nanoparticle transport [21]. These properties, in combination with the immunostimulant nature of bacteria, make this strategy a powerful tool to induce sustained and effective immune responses against tumors. In this work, the recent advances achieved in the development of bacteria as nanoparticle carriers for antitumor immunotherapy are presented to describe a powerful strategy to induce sustained and efficient immune responses against tumors. 

## 2. Nanoparticles for Immunotherapy

The establishment of a solid tumor follows a complex three-stage process known as the three Es: elimination–equilibrium–escape [22]. At the beginning, pro-tumor mutated cells are rapidly recognized and killed by immune cells (elimination stage). After a certain time, these mutated cells can acquire the capacity to partially avoid the attack of immune cells entering the equilibrium stage. In this stage, the tumoral cells remain in a dormant state for long periods of time—even decades. Finally, in some cases, cancerous cells acquire a set of capacities that allow them to escape from the control of the immune system (escape phase). At this point, immune cells are no longer a barrier to the development of the tumor, and can even become an ally of the malignancy, inducing the formation of tumoral blood vessels, excreting proteolytic enzymes that degrade the extracellular matrix, or maintaining the immunosuppressive environment [23]. An efficient antitumor response by the immune system follows a stepwise process known as the cancer-immunity cycle [24]. Tumoral cells present large number of mutations and, therefore, they express mutated proteins that can be recognized by immune cells [25]. These tumoral antigens are captured by immature antigen-presenting cells (APCs), which are present in the neoplastic tissue. In normal conditions, the presence of danger signals produced by the tissue triggers APC maturation [26]. Mature APCs present the tumoral antigen on their membranes, and migrate to lymph nodes where they bind to naïve T lymphocytes endowed with specific receptors against the antigens of the tumor triggering their clonal selection and activation. Active T cells enter the bloodstream and travel to the tumoral tissue, where they locate and destroy the cancerous cells. This tumoral cell destruction releases more tumoral antigens, continuing the cycle until the complete eradication of the malignancy. During the equilibrium stage, tumoral cells begin to develop the capacity to surpass the different steps of this cycle. The goal of immunotherapy is to reboot the cycle, resetting the natural capacity of immune cells to fight against tumors. In recent years, nanoparticles have become an excellent platform to deliver immunotherapeutic agents, due to their tendency to interact with immune cells. Among the different types of immune cells that comprise the immune system, two groups can be differentiated: innate and adaptive immune cells. Nanomedicines loaded with immunotherapeutic agents have been engineered to interact with each cell subpopulation, with the aim of triggering specific antitumor responses (Figure 1) [27].

Nanoparticles designed to interact with cells of the innate immune system aim to induce the maturation of APCs endowed with tumoral antigens, which can be transported by the nanocarriers (nanovaccination) or can be generated in the tumoral tissue by the action of the nanocarrier (in situ vaccination). One of the most widely used strategies is the use of nanoparticles as selective carriers to deliver tumoral antigens to APCs in combination with adjuvants, as specific nanovaccines against the tumor [28]. The intramuscular administration of these nanovaccines provokes tissue damage, which induces the infiltration of APCs into the zone. Once there, APCs capture the nanocarrier through endocytosis. The combination of the release of tumoral antigens and adjuvants, which are usually molecules that bind to cell receptors that recognize pathogen- and danger-associated molecular patterns (PAMPs and DAMPs, respectively), triggers APC maturation. Mature APCs migrate to lymphoid organs, where they induce the clonal selection of specific lymphocytes against the tumor. As a representative example of this strategy, Kuai et al. reported the use of nanodiscs composed of synthetic high-density lipoprotein (sHDL) decorated with tumor antigens of MC-38 colon carcinoma and CpGs as adjuvants, inducing the proliferation of specific CD8+ T lymphocytes [29]. CpGs are unmethylated oligonucleotide chains enriched in guanidine (G) and cytosine (C), and are usually present in viruses, bacteria, and parasites. Therefore, APCs recognize these CpGs through the specific Toll-like receptor TLR9, inducing their maturation. Another interesting approach is to synthesize the nanocarriers by employing the cell membrane of the tumoral cells to exploit the complete set of tumoral antigens [30]. Additionally, the tumoral membrane can be employed as a coating for adjuvant-loaded nanoparticles to induce specific adaptive immune responses [31]. These strategies present some important limitations—on the one hand, it is necessary to identify and isolate suitable tumoral antigens, which is a complex and time-consuming process; on the other hand, the expression of these tumoral antigens presents large variation between patients, and even between the different metastatic lesions within the same patient [32]. This fact means that the efficacy of the therapy can present significant variations in each patient. One interesting approach is in situ vaccination. This strategy is based on the administration of chemotherapeutic drugs, immunostimulants, and/or thermal/radiation therapy to induce immunogenic cell death (ICD) of the tumoral cells, which triggers an efficient immune response [33]. ICD is a particular type of cell death in which the dying cells have “eat me signals” on their membrane—such as calreticulin (CRT)—which induce their phagocytosis by APCs, and release “danger signals” such as adenosine triphosphate (ATP), high-mobility group protein B1 (HMGB1), or heat-shock proteins (HSPs), which stimulate APC maturation [34]. Herein, the tumoral antigen source is the tumor itself; therefore, the antigen isolation step is not required. Additionally, the same strategy can be applied to different patients, because each of them will produce their own tumoral antigens. This strategy allows the training of the immune system to fight against tumors, which are in continuous evolution and present different antigens over time. ICD can be provoked by different insults, such as cytotoxic drugs (e.g., mitoxantrone, anthracyclines, oxalilplatin, cyclophosphamide, bortezomib), UV radiation, photodynamic therapy (PDT), or radiotherapy [35]. Lu et al. reported the use of hybrid nanocarriers composed of mesoporous silica nanoparticles coated with lipid bilayers as ICD inducers in an animal model of pancreatic ductal adenocarcinoma [36]. In this nanodevice, oxalilplatin was loaded within the silica pore network, while indoximod—an inhibitor of indoleamine 2,3-dioxygenase 1 (IDO1)—was retained in the lipid bilayer. IDO1 is regionally expressed by pancreatic cancer cells, catalyzing the conversion of L-tryptophan (Trp) to L-kynurenine. The reduction in the availability of Trp hampers the development of cytotoxic T cells and induces the proliferation of regulatory T cells (T_regs_), which maintain an immunosuppressive environment in the diseased tissue [37]. The simultaneous release of oxalilplatin and indoximod triggers a potent ICD of the pancreatic tumoral cells, followed by a reduction in the immunosuppressive tumoral microenvironment, which leads to recruitment of cytotoxic CD8+ T lymphocytes, depletion of T_regs_, and significant tumor shrinkage. Wang et al. synthesized dendrimers conjugated with catechols and RGD peptides as targeting moieties to deliver bortezomib—a proteasome inhibitor approved for the treatment of different malignancies—to metastatic bone tumors [38]. RGD moieties bind to αβ-integrin receptors usually overexpressed by tumoral cells, enhancing the dendrimer uptake within the diseased cells. Bortezomib was loaded via catechol–boronate linkage, which presents pH-responsive behavior, suffering cleavage under mild acidic conditions (Figure 2). Thus, the nanodevice can be selectively engulfed by tumoral cells via endocytosis, and then the acidic environment present in the endosomes triggers bortezomib release via hydrolysis of the boronate bonds.

The local temperature increase promoted by induced hyperthermia provokes a significant cellular stress, which results in overexpression of HSPs, which are danger signals that stimulate APC maturation [39]. Beola et al. studied the application of magnetic nanoparticles for immunotherapy in the treatment of pancreatic cancer [40]. In this work, poly(maleic anhydride-alt-1-octadecene) (PMAO)-coated iron oxide nanoparticles were intratumorally injected in a murine xenograft model of pancreatic cancer, showing higher nanoparticle penetration when an alternative magnetic field was applied—probably due to the disruption of the extracellular matrix. In all of the mice treated with magnetic nanoparticles, significant tumor shrinkage and high expression of calreticulin were observed when alternative magnetic fields were applied. Nanoparticles have been loaded with photodynamic therapy agents, which are molecules that produce reactive oxygen species (ROS) that induce ICD under light irradiation [41]. Choi et al. developed self-assembled nanoparticles composed of a photosensitizer (verteporfin) and a cytotoxic drug (doxorubicin) linked with a cathepsin-B-cleavable peptide [42]. This nanodevice was accumulated in lung tumors via the EPR effect and, once there, the presence of cathepsin B—which was overexpressed in the malignancy—triggered the release of doxorubicin and verteporfin in the tissue. The synergic effect between doxorubicin and the ROS produced by the irradiation of verteporfin with visible light induced potent immune activation responses. Chen et al. developed core–shell PLGA nanoparticles loaded with catalase and imiquimod, in order to enhance the efficacy of radiotherapy in immunotherapy [43]. The role of catalase in this system was to restore normoxia in the tumoral tissue through the catalytic transformation of hydrogen peroxide present in the zone into molecular oxygen, which enhanced the ICD induction capacity of radiation. 

Nanoparticles can be designed to interact with adaptive immune cells to enhance or facilitate the natural capacity of these cells to orchestrate the selective eradication of the tumoral cells—not only in primary tumors, but also in distant metastatic lesions. Thus, bispecific nanoparticles have been decorated with two different types of antibodies to act as bridge between cytotoxic T cells and tumoral cells [44]. In this work, liposomes were functionalized with anti-CD3 monoclonal antibodies that bind to T cells and anti-CD20 antibodies that were used to target Waldenström’s macroglobulinemia (WM) tumoral cells. These nanoparticles presented a longer half-life in the blood than free antibodies, and induced 60–70% tumoral cell elimination in the presence of T cells in tridimensional tumor models. Intravenous administration of these nanodevices and humanized T cells in a murine MW xenograft model yielded complete tumor eradication by day 35. Additionally, the complete cohort of mice treated with the nanoparticles and T cells did not show signs of disease for 2 months after the experiment stopped. Magnetic nanoparticles have been used to guide T cells to malignant tissues via the application of external magnetic fields [45]. Thus, amino-functionalized iron oxide nanoparticles were effectively engulfed by Jurkat and murine T cells. Intravenous administration of T cells loaded with these magnetic cores in a murine model, followed by the application of an external magnetic field in popliteal lymph nodes, induced significant T-cell accumulation in the zone. The immunosuppressive environment usually present in solid tumors can be characterized by different hallmarks, including the presence of enzymes that deplete compounds involved in T-cell function, accumulation of immunosuppressive cells such as tumor-associated macrophages (TAMs) or T_regs_, and the expression on the tumoral cell membrane of specific ligands that trigger T-cell apoptosis, such as the programmed death ligand 1 (PD-L1) [46]. T_regs_ play a paramount role in the maintenance of the immunosuppressive environment, and different nanoparticle-based strategies have been developed to eliminate this cell population in the tissue, from the use of magnetic hyperthermia [47] to the controlled delivery of specific drugs such as imatinib to hamper their function [48]. Similarly, TAMs are the target of different nanomedicines that aim to deplete them or to reprogram their function from the pro-tumor M2 phenotype to the antitumor M1 phenotype [49]. Polymeric micelles have been employed to deliver immunomodulators such as imiquimod to stimulate the maturation of macrophages, restoring their antitumor capacity [50]. Nanoparticles endowed with anti-PD-L1 were able to block the PD-L1 receptors located on the membranes of tumoral cells, enabling their destruction by activated T-effector cells [51]. Polymeric nanoparticles functionalized with folic acid to target tumoral cells were loaded with doxorubicin—as an ICD inducer—and microRNA to knock out PD-L1 expression [52]. The administration of these nanocarriers in murine tumoral models showed a significant inhibition in the tumoral growth, increased dendritic cell maturation, and high infiltration of CD8+ T cells in the tumoral tissue.

Despite the great advances carried out in the development of nanoparticle-based immunotherapies, some biological barriers remain, hampering their efficacy. It is crucial to enhance the nanoparticles’ localization in the tissue, and to achieve a homogeneous distribution in the diseased zone, in order to concentrate the therapeutic effect in the required zone and to reduce the side effects associated with the administration of immunomodulators. As mentioned in the Introduction, ultrasound has been used to enhance the penetration of nanoparticles in living tissues [15], but it can be also employed as an ICD inducer due to the thermal tissue damage that can be induced by its application [53]. The used of living organisms to transport these nanomedicines can provide a powerful tool to improve the efficacy of immunotherapy due to the combination of chemotaxis and self-motility of bacteria and cells with their own immunostimulant properties, which can provide a synergic action that is difficult to achieve by other means.

## 3. Bacteria for Antitumor Therapy

One of the first cases that employed bacteria for cancer treatment was described in 1898 [54]. Dr. William Coley noticed that the existence of bacterial infections was strongly correlated with tumor growth shrinkage. He observed that maintaining a balance between infection control and therapeutic effect was crucial for successful therapy. For this, a combination of heat-inactivated *Serratia marcescens* and *Streptococcus pyogenes*—called “Coley’s toxin”—was administered to many patients with inoperable soft-tissue sarcomas [55,56]. The treatment was very effective, and Coley’s vaccine was tested on more than 1000 cancer patients. 

However, bacteria-mediated tumor therapy (BMTT) was no longer used after Coley’s death. Anticancer therapy has been based, for decades, on the administration of highly cytotoxic drugs for the inhibition of cell proliferation or to promote cancer cell death [57]. Nevertheless, this strategy focuses on rather non-specific cell targets that are not restricted to malignant cells, but are also common to every healthy cell, such as DNA repair mechanisms or topoisomerase inhibitors. The poor selectivity of chemotherapeutic agents and the non-specific distribution throughout the body generally provoke severe damage in healthy tissues or organs, as well as impairment in the immune systems of patients [58]. Moreover, the poor diffusion of drugs towards the inner regions of tumoral tissues is closely related to the early appearance of multidrug resistance (MDR) mechanisms. MDR strongly limits the therapeutic efficacy of chemotherapy, and is commonly credited for increased morbidity and mortality [59]. 

Increased understanding of the pathophysiology and immunology of cancer in recent decades has shown Coley’s observations to be valid, demonstrating that most cancers are very sensitive to a boosted immune system resulting from immunostimulation [60]. Consequently, antitumor strategies have turned the spotlight on the activation and reprogramming of the immune system for patients to fight cancer by themselves, through the employ of immunostimulatory agents within biological therapies [61,62]. Biological therapies are generally more effective and physiologically well-tolerated than conventional chemotherapy, resulting in prolonged survival of cancer patients with poor prognosis [63]. The success of immunotherapies is mainly based on two facts: (1) they induce a memory function of the adaptive immune system [64]—in contrast to chemotherapy, which destroys cells—and (2) they specifically activate the immune system against cancer; thus, the side effects are low, due to autoregulation of immunotolerance mechanisms to self-antigen mechanisms [65,66]. However, as discussed in previous sections, tumoral cells are known to reprogram immune cells to evade immunosurveillance via a variety of mechanisms. These include the secretion of immunosuppressive factors that induce apoptosis of lymphocytes, alterations of the T-cell antigen-presentation machinery, or activation of negative regulatory pathways that induce immunotolerance [67,68]. Moreover, immunotolerance mechanisms produced by tumoral cells may also promote the apparition of MDR mechanisms in response to biological therapies [69,70]. An insufficient immune activation against tumors is the main reason that leads to treatment failure. 

In this context, the use of BMTT is re-emerging as a powerful tool to combat cancer [71]. The use of bacteria presents many advantages compared with biological therapies, since they possess unique features that allow them to overcome most of these shortcomings [72,73]. Bacteria can (1) selectively colonize and replicate within the tumor microenvironment, guided by oxygen deficiency [74]. Hypoxia is a unique characteristic of tumoral tissues, and is not present in other organs of the healthy body. This fact implies that bacteria can specifically target and accumulate in the anoxic and necrotic regions of tumors [75], and this predisposition differs depending on their oxygen tolerance. Strict anaerobic bacteria (such as *Clostridium* and *Bifidobacterium*) cannot tolerate oxygen; thus, they can only survive and proliferate in deep hypoxic areas of tumors [76]. On the other hand, facultative anaerobic bacteria (as *E. coli* or *Salmonella*) can tolerate oxygen, but preferentially accumulate in hypoxic regions. 

Moreover, inherent bacterial immunogenicity (2) represents an advantage to fight cancer. This fact gives bacteria potent immunostimulatory capacity, as bacterial infection activates the innate immunity by recruiting neutrophils and CD8+/CD4+ T lymphocytes towards the tumoral tissue for bacteria eradication [77]. Immune cell infiltration, together with the release of cytokines and chemokines, produces potent local immunostimulation with no effect on the surrounding healthy tissue [77]. In addition, bacterial active motion capacity (3), thanks to their flagella, is a clear advantage. They can navigate upstream of the interstitial fluid pressure (IFP) and the dense fibrotic tumoral matrix, allowing them to reach deeper regions of the tumoral tissue that are unachievable with conventional therapies. Moreover, genetic manipulation of bacteria (4) can also enable the expression of multiple cytotoxic agents, cytokines, immunomodulators, immune checkpoint nanobodies, or antitumor antigens that increase the therapeutic effect against malignancies [78]. Synthetic biology emerges in this case as a powerful tool for the in situ release of therapeutic payloads in the tumoral microenvironment [79]. The wide range of biotherapeutics that can be produced can be rationally used for combination therapies. 

Bacteria can thus become recombinant factories that express immunomodulators, with the aim of stimulating antitumor responses and leukocyte migration toward tumoral tissues to promote tumor regression. Once bacteria have colonized deep regions of the tumor, gene expression is triggered, and the bacteria can produce the therapeutic agent in high amounts. This results in a significant accumulation of the released payload in deep regions of the tumor for longer periods of time, enhancing the therapeutic outcome. Some examples include the heterologous expression of cytokines such as IL-2 [80] or immunomodulatory proteins such as flagellin B [81]. Another strategy is the administration of probiotic bacteria engineered for the controlled production and intratumoral release of nanobodies that target programmed cell death ligand 1 (PD-L1)—an important immune checkpoint inhibitor [82]. Another example is the use of attenuated *Salmonella* as a DNA vaccine that targets VEGFR2, which induces tumor vessel collapse by triggering a T-cell-mediated immune response against proliferating endothelial cells [83]

A complementary strategy for bacteria-mediated gene expression is gene silencing. Gene silencing can be accomplished by transferring plasmids encoding short hairpin RNA (shRNA) that produce small interfering RNA (siRNA). Silencing of the immunosuppressant protein indoleamine 2,3-dioxygenase (IDO) with shRNA plasmids that target the gene encoding IDO, or silencing of STAT3 with shRNA-loaded *Salmonella* [84], are two examples. Silencing of IDO expression leads to increased neutrophil infiltration and intratumoral cell death [85].

The biological interactions established between cancer cells, bacteria, and the surrounding tumor microenvironment provoke complex alterations in tumor-infiltrated immune cells and in chemo/cytokines which, in turn, promote tumor regression [72]. Tumor regression can be induced by several different mechanisms once bacteria have colonized and grown within the tumoral tissue. Depending on the bacterial strain, the display of tumor suppression mechanisms within the tumor microenvironment is different. One mechanism is the in situ production of bacterial toxins of *Salmonella*, *Listeria*, or *Clostridium*, which directly kill tumor cells by inducing apoptosis or autophagy [86]. For example, *Salmonella* toxins induce upregulation of the ubiquitinated protein connexin 43 in tumor cells, promoting the formation of junctions between malignant cells and dendritic cells. These functional connections allow cross-presentation of tumor antigens to dendritic cells, leading to reduced expression of the immunosuppressive IDO enzyme in T cells, and a consequent and specific increase in CD8+ T-cell activation [87]

Concerning the host–pathogen interaction, the rapid proliferation of bacteria in tumoral tissues and the presence of bacterial components such as lipopolysaccharide (LPS) and flagellin induce significant migration of innate immune cells—such as macrophages, neutrophils, and dendritic cells—towards colonized tumors. Then, inflammasome activation leads to a strong production of interleukin-1β by macrophages and dendritic cells [88]. LPS is involved in the high secretion of the TNF-α cytokine via interactions with CD14 and TLR4 receptors [89]. *Salmonella* flagellin is involved in inflammasome-driven secretion of interleukin-1β and interleukin-18, which serve as activators of IFN-γ—a cytokine that induces the production of cytotoxic T cells and natural killer (NK) cells [90]. These unique characteristics of bacteria for cancer treatment are unachievable with both conventional chemotherapy and biological therapies. The major limitation of BMTT is the pathogenicity of most strains, but the safety profile of bacteria can be improved via the deletion of major virulence genes through genetic engineering, which enables the suppression of pathogenic-related genes [91]. *C. novyi*-NT—a non-pathogenic *Clostridium* clone—is commonly used in BMTT, and is obtained by deleting the gene encoding the α-toxin—the major toxin responsible for their virulence [92]. In *Salmonella*, the elimination of the msbB gene strongly reduces the toxicity of bacteria by 10,000-fold [91]. 

However, bacterial motility can additionally be exploited to enhance the therapeutic effect of immunotherapies [93]. These motile microorganisms can also serve as microswimmers for the delivery of drug-loaded nanoparticles within a living organism [20,94,95]. Diverse biohybrid nanocarriers have been prepared for use in multiple therapeutic strategies [96]. The obtention of these nanosystems involves the attachment of either single molecules [97] or a broad variety of nanomaterials of different natures onto the bacterial surface. Biohybrid carriers of inorganic nanoparticles—such as gold nanoparticles [98,99] or mesoporous silica nanoparticles (MSNs) [100]—are some examples. Liposomal [101,102,103] or polymeric [104,105,106,107,108] nanoparticles are also widely employed for cargo delivery assisted by bacteria. If we consider the diversity of existing nanomaterials, this capacity of bacteria is particularly relevant. Biohybrid nanosystems can be engineered for the transport of high amounts of diverse therapeutic agents (drugs, proteins, nucleic acids) within nanoparticles (Figure 3). These nanocarriers can produce synergistic effects when engineered bacteria are used for the secretion of immunomodulators, achieving a targeted delivery to hypoxic regions of tumors and a precise therapy that greatly enhances the therapeutic result.

Moreover, anchoring of nanoparticles on the surface of bacteria does not significantly affect their motility and viability [109,110,111]. Binding between bacteria and nanoparticles can be easily achieved through electrostatic interactions, [112], by establishing biotin-based bioaffinity interactions [104], or by employing bioconjugation reactions [100,113,114], among other methods. Overall, the required properties for bacteria to become appropriate nanocarriers are high motion capacity and tumor chemotaxis. The motility is provided by the flagella, while chemotaxis can be provided by different characteristics present in tumors, such as the presence of hypoxia, or tumor-related metabolites such as lactic acid excess. Thus, only flagellated and hypoxia- or nutrient-attracted microorganisms are useful as nanocarriers in cancer therapy. Attenuated strains of *E. coli*, *Salmonella*, or *Clostridium* fulfill these essential requirements, so the works described below mainly used these bacteria as effective carriers.

## 4. Bacterial Nanocarriers for Cancer Immunotherapy

Immunotherapy-based strategies to fight cancer are gaining more and more importance. In recent years, several works using bacterial nanocarriers for immunomodulation have been presented. Here, we describe some of these works based on the different approaches employed.

### 4.1. Vectors of Oral DNA Vaccines

One of the first examples in the use of bacteria for cancer immunotherapy was their employ as vectors of DNA vaccines [115]. Oral administration of bacteria allows the colonization of the gut-associated lymphoid tissue [116]. This promotes a strong and enduring immunological response, since the bacteria are then recognized and eliminated by resident macrophages and dendritic cells present in the tissue. Hu et al. designed a cationic nanoparticle-coated vehicle based on live attenuated *Salmonella* that efficiently delivered an oral DNA vaccine [108]. The cationic polymeric nanoparticles were designed to fulfill the two main requirements in oral delivery: the polymeric coating of the bacteria provides an enhanced tolerance to the acidic pH in the stomach, while the polycationic nature of this coating facilitates the efficient phagosomal escape of bacteria from immune cells. For this, they coated bacteria with a nanoparticle assembly of cationic β-cyclodextrin-PEI600 polymers and plasmid DNA encoding vascular endothelial growth factor receptor 2 (VEGFR2). This hybrid nanosystem exhibited remarkable T-cell activation and cytokine production. Successful oral delivery of VEGFR2 also promoted efficient inhibition of tumoral growth and tumor necrosis in a murine B16 melanoma model, due to the suppression of angiogenesis in the tumoral vasculature. 

Naciute et al. also developed a biohybrid-based vaccine as an oral delivery system for colorectal cancer [117]. For this, *E. coli* bacteria were attached to tumor antigen-containing and adjuvant-containing liposomes either by electrostatic interactions between negatively charged *E. coli* and cationic liposomes, or by using streptavidin-decorated liposomes and biotin-functionalized *E. coli*. Biohybrid vaccines demonstrated potent in vitro and in vivo therapeutic effects, and significantly heightened the expression of CD80+, CD40+, and CD86+ surface proteins on murine bone-marrow-derived dendritic cells. Mice vaccinated with biohybrid vaccines had increased CD8+ T-cell infiltration into the diseased tissue, and developed threefold smaller tumors compared with the control vaccine without *E. coli*.

### 4.2. Agents for Macrophage Polarization

Another example of immunomodulation mediated by bacterial nanocarriers is that reported by Wei et al. [118], who designed a biohybrid system composed of PLGA nanoparticles attached to the surface glycol of chitosan-decorated non-pathogenic *Escherichia coli* (Ec-PR848) via electrostatic absorption for a combination cancer immunotherapy (Figure 4a). Two different PLGA nanoparticles were prepared by loading with both the Toll-like receptor (TLR) 7/8 agonist resiquimod (PR848) and doxorubicin (PDOX). DOX is a well-known drug that induces ICD by promoting the maturation of dendritic cells and activating and recruiting cytotoxic T lymphocytes in tumors [119]. On the other hand, resiquimod promotes efficient polarization of TAMs from the M2 to the M1 phenotype [120], boosting the secretion of inflammatory cytokines and, thus, activating antineoplastic immunity and avoiding immunosuppression.

The tumor-hypoxia-targeting capacity of Ec-PR848 resulted in a 17-fold greater accumulation of bacterial colonies within tumoral tissue compared with the kidneys (Figure 4b). The biohybrid nanosystem was phagocytosed by M2 TAMs, and effectively transformed M2 macrophages into activated M1 macrophages with an M1/M2 ratio of 1.34. When combining both treatments (Ec-PR848 and PDOX), Ec-PR848 was able to enhance the polarization effect, with an M1/M2 ratio of 1.59. This strategy exhibited that low-dose chemotherapy-induced ICD combined with TAM polarization therapy can significantly enhance the efficacy of immunotherapy for tumor growth suppression by improving the infiltration of cytotoxic T lymphocytes in tumor tissues and activating antitumor immunity.

Ektate et al. also designed a nanocarrier for macrophage polarization using attenuated *Salmonella* that actively transports low-temperature-sensitive liposomes (LTSLs) inside colon cancer cells [121]. The aim of these “thermobots” was the simultaneous triggering of doxorubicin release and polarization of macrophages to the M1 phenotype with high-intensity focused ultrasound heating (40–42 °C). The thermobots exhibited efficient intracellular trafficking, elevated nuclear localization of doxorubicin, and induced pro-inflammatory cytokine expression in colorectal cancer cells. Combination therapy with thermobots and ultrasound heating (~30 min) significantly enhanced the therapeutic efficacy and polarization of macrophages to the M1 phenotype in murine colon tumor models. 

### 4.3. Providers of Bacterial Metabolites for Immunostimulation

An innovative strategy for immune system activation is the work reported by Chen et al. [122], who leveraged the ability of *Shewanella oneidensis* MR-1 to reduce the number of metallic ions for the anaerobic catabolism of lactate through the transfer of electrons to metallic nanoparticles. For this, a lactate-fueled biohybrid nanosystem (Bac@MnO_2_) was prepared by incorporating MnO_2_ nanoflowers on the bacterial surface. MnO_2_ nanoflowers act as electron receptors, while lactate—a tumor metabolite—acts as an electron donor to construct a bacterial respiration pathway at the tumor site. This redox system results in the continuous catabolism of intercellular lactate, which suppresses tumor growth by exhausting lactate within the tumor. Lactate plays a critical role in tumor development. Apart from serving as an energy source for the tumor, lactate can also weaken immunosurveillance of T cells and NK cells by inhibiting the production of cytokines such as IFN-γ [123]. After Bac@MnO_2_ treatment, a high IFN-γ concentration was detected compared to other samples, which indicates that low intratumoral lactate concentrations could induce significant immune activation, contributing to tumor regression.

A similar anticancer strategy involving a cancer metabolite is the in situ generation of cytotoxic species by photoinduced reduction of NO^3−^ to NO. Zheng et al. designed a biohybrid system by anchoring *E. coli* bacteria with a photocatalyst of carbon nitride (C_3_N_4_) doped with carbon dots that enhance the metabolic activity of bacteria [112]. *E. coli* bacteria contain enzymes that reduce endogenous NO^3−^ to cytotoxic NO. Under light irradiation, photoelectrons are transferred from C_3_N_4_ to *E. coli* to promote the enzymatic reduction of NO^3−^, resulting in a 37-fold increase in intratumoral NO concentrations. C_3_N_4_-loaded bacteria accumulated throughout the tumor in a murine model, and the treatment resulted in about 80% inhibition of tumor growth. They also found that photocontrolled bacterial metabolite therapy induced immunostimulation, since immune activation of both dendritic cells and CD8+ cytotoxic T cells was observed, along with increased concentrations of immune cytokines in serum. 

### 4.4. Carriers for Immunogenic PDT/PTT

Another strategy for tumor regression is the use of combination therapies combining PDT with immunostimulation. In the work reported by Liu et al. [124], the photosynthetic bacterium *Synechococcus* 7942 was employed for the delivery of a photosensitizer to the tumoral mass for in situ oxygen generation to achieve photosynthesis-boosted PDT. For this, human serum albumin nanoparticles were loaded with the photosensitizer indocyanine green (HSA/ICG), and they were attached to the surface of bacteria by amide bonds to form the nanosystem S/HSA/ICG. Once injected intravenously into tumor-bearing mice, S/HSA/ICG effectively accumulated in the tumor and generated oxygen when irradiated with a 660 nm laser. The continuous production of oxygen remarkably ameliorated tumor hypoxia and promoted reactive oxygen species production, thereby inducing tumor cell death. Moreover, this nanocarrier could also effectively revert the tumor’s immunosuppressive microenvironment, and increased antitumor immune responses by generating IL-10, increasing IFN-γ production, and inhibiting production of the immunosuppressive mediator TGF-β. S/HSA/ICG showed an optimal effect in preventing tumor recurrence and inhibiting metastasis in a murine model of metastatic triple-negative breast cancer. This work represents an interesting approach for enhancing immunogenic PDT with photosynthetic bacteria for tumor remission.

The combination of immunotherapy with additional functions, such as PDT or photothermal therapy (PTT), allows the generation of synergistic effects that enhance the therapeutic outcomes. Similarly to the previous work, Chen et al. designed a nanosystem [125] for a triple therapy based on biocompatible polydopamine nanoparticles coating the surface of anaerobic *Salmonella* VNP20009. The coated nanosystem was intravenously injected, followed by irradiation with a near-infrared laser at the tumor zone, and combined with local administration of a phospholipid-based gel loaded with the anti-PD-1 peptide AUNP-12. This gel sustainably releases the AUNP-12 peptide for 42 days, thus maintaining the immunopermissive tumor microenvironment. The triple combination of PTT, biotherapy, and PD-1 blockade showed high efficacy in achieving strong antitumor immune responses and eliminating large tumors in 50% of animals from a murine model of melanoma over the course of 80 days. These results yield new insights about the advantages of BMTT, and this innovative strategy may represent a powerful anticancer immunotherapy tool.

By combining the properties of bacteria depicted herein with multiple nanoparticle formulations, two or more therapeutic approaches can be incorporated into a single nanosystem capable of producing excellent therapeutic effects. Therefore, the advancement of bacteria-based immunotherapy has greatly developed in recent years, and offers the prospect of being a very powerful tool for cancer treatment in the coming years.

## 5. Conclusions

Despite the huge efforts carried out by the scientific community in recent decades towards the development of smarter and safer nanocarriers, their routine implantation in clinical oncology is still far off. The existence of strong biological barriers compromises their performance. The conventional dogma of the passive accumulation of nanoparticles in solid tumors is subject to intense debate. Nanoparticle accumulation in malignant tissues is highly dependent on the tumor type, patient, and disease stage. Additionally, poor penetration of nanoparticles in solid malignancies causes their inhomogeneous distribution, which hampers their capacity to eradicate the tumor. Therefore, oncological nanomedicine is now at a crossroads. In recent years, the nanomedicine community has shifted the target from the tumoral cell to immune cells in order to exploit the natural tendency of the immune system to interact with nanometric bodies. Immune cells have evolved over the course of millennia to recognize pathogens in the nano- and micrometric range. Thus, different nanodevices have been engineered, with the aim of triggering antitumor immune responses. The use of microorganisms such as bacteria as nanoparticle carriers combines the chemotaxis and self-motility of living organisms with their own immunomodulatory nature. This field, still in its early stages, could provide valuable strategies to deliver immunotherapeutic agents to poorly irrigated tumors, and open a new way to fight against these complex diseases.

## Figures and Tables

**Figure 1 pharmaceutics-14-00784-f001:**
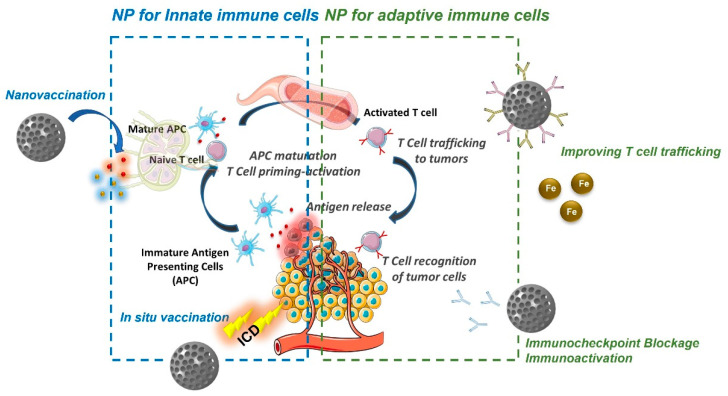
Nanomedicine-based strategies to reboot the cancer-immunity cycle. Figure Reproduced from Ref. [27], Molecules, 2020.

**Figure 2 pharmaceutics-14-00784-f002:**
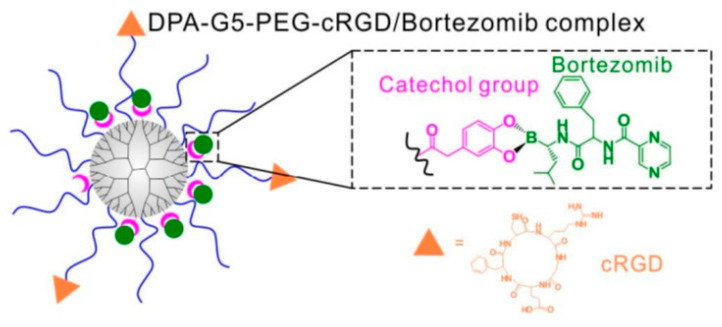
Dendrimers decorated with RGD patterns and catechol groups for bortezomib attachment by catechol–boronate linkage. Figure reproduced with permission from Ref. [38], ACS Appl. Mater. Interfaces, 2018.

**Figure 3 pharmaceutics-14-00784-f003:**
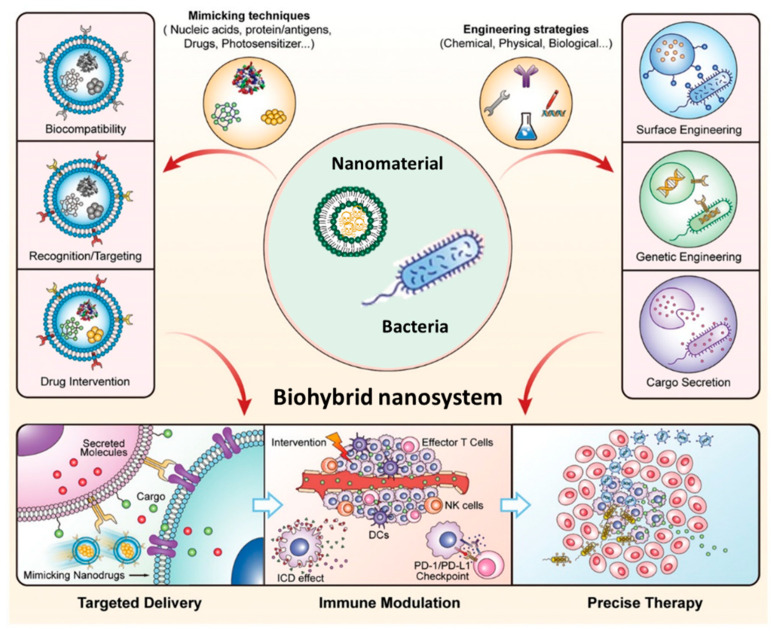
Schematic summary of the main characteristics of BMTT and the different approaches for the obtention of biohybrid nanomaterials based on bacteria for the targeted delivery of payloads, immune system modulation, or precise therapy against cancer. Reproduced with permission from Ref. [93], Advanced Materials, 2021. Copyright © 2021, Wiley -VCH GmbH.

**Figure 4 pharmaceutics-14-00784-f004:**
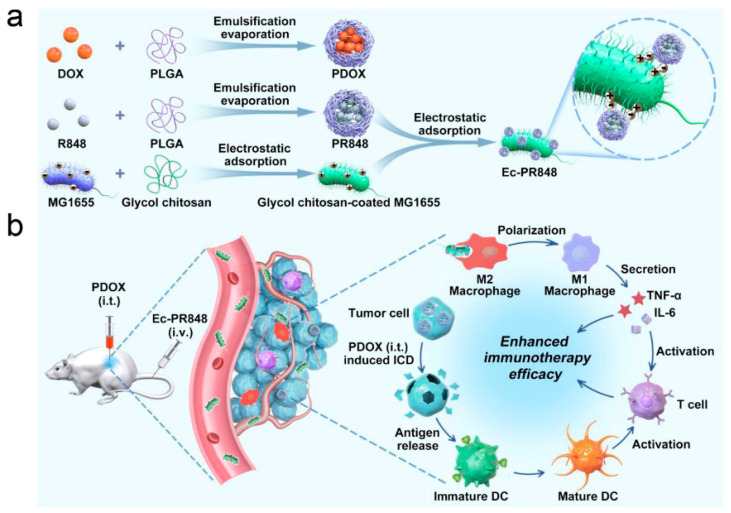
Scheme for (**a**) the preparation of resiquimod-loaded PLGA–*E. coli* (Ec-PR848) in combination with PDOX; (**b**) PDOX-induced ICD enhances the immunotherapeutic effect derived from TAM polarization. Repoduced with permission from Ref. [118], Nano Letters, 2021. Copyright© 2021, American Chemical Society.

## Data Availability

Not applicable.

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
