# Peer review of "Bacteria as Nanoparticle Carriers for Immunotherapy in Oncology"

_pharmaceutics, 2022, doi:10.3390/pharmaceutics14040784_

Round 1

Reviewer 1 Report

The manuscript comprehensively summarized applications of bacteria as nanoparticle carriers for tumor immunotherapy, which is gaining more and more attention owing to the unique characteristics of bacteria. Overall, the topic is interesting. Below are a few comments that are suggested for consideration during its revision.

  1. In the part Nanoparticles for immunotherapy, a great deal of space is devoted to the application of PDT nanoparticles in anti-tumor immunity. As the protagonist of this paper is bacteria, it is appropriate to simplify the content of this aspect and expand the description of immunotherapy combined with radiotherapy, ultrasound therapy, etc., to compare its advantages and disadvantages with bacterial nanocarriers.
  2. The manuscript follows a clear thread from the immunotherapy of nanomaterials to the application of bacterial nanocarriers in anti-tumor immunity. Here's a suggestion that setting subheadings for each of the different types or functions of bacterial nanocarriers, which would make the text clearer and easier to read.

Author Response

The comments of this referee are deeply acknowledged since they help to improve the quality of the manuscript. In accordance with his/her main suggestions the following points have been addressed:

1) In the part Nanoparticles for immunotherapy, a great deal of space is devoted to the application of PDT nanoparticles in anti-tumor immunity. As the protagonist of this paper is bacteria, it is appropriate to simplify the content of this aspect and expand the description of immunotherapy combined with radiotherapy, ultrasound therapy, etc., to compare its advantages and disadvantages with bacterial nanocarriers.

According with the reviewer suggestion some paragraphs which addressed the application of PDT in antitumoral therapy has been removed or simplified from the manuscript in order to simplify this section. Additional paragraphs have been introduced in the manuscript to describe the use of ultrasounds and radiotherapy have been added to the manuscript: (Page 5) “Chen et al. have developed core-shell PLGA nanoparticles loaded with catalase and imiquimod, in order to enhance the efficacy of radiotherapy in immunotherapy.[43] The role of catalase in this system was to restore the normoxia in the tumoral tissue through the catalytic transformation of hydrogen peroxide present in the zone into molecular oxy-gen which enhanced the ICD induction capacity of radiation.” and   (Page 6) “As it has been mentioned in the introduction, ultrasounds have been used to enhance the penetration of nanoparticles in living tissues[15] but they can be also employed as ICD inducers due to the thermal tissue damage which can be induced by their application.[53] The used of living organisms to transport these nanomedicines would provide a powerful tool to improve the efficacy of immunotherapy due to the combination of chemotaxis and self-propelled skills of bacteria and cells with their own immunostimulant properties which can provide a synergic action hardly achieved by other ways.”  Please, note that Refs numeration have been corrected in the new manuscript. The extension of these new paragraphs has been limited to not increase the length of this section.

2) The manuscript follows a clear thread from the immunotherapy of nanomaterials to the application of bacterial nanocarriers in anti-tumor immunity. Here's a suggestion that setting subheadings for each of the different types or functions of bacterial nanocarriers, which would make the text clearer and easier to read.

According with the reviewer suggestion, we have included the subsections 4.1, 4.2, 4.3 and 4.4 in order to organize the bacterial nanocarriers described in the manuscript and to facilitate its easy location in the section. Thus, the next lines have been added:

Lines 416-419: “Immunotherapy-based strategies to fight cancer are gaining more and more importance. In recent years, several works using bacterial nanocarriers for immunomodulation have been presented. Herein we describe some of these works based on the different approach employed.

4.1. Vectors of oral DNA vaccines”

At line 446: “4.2. Agents for macrophages polarization”

At line 481: “4.3. Providers of bacterial metabolites for immunostimulation”

At line 508: “4.4. Carriers for immunogenic PDT/PTT”

Reviewer 2 Report

  1. There are some grammatical mistakes in the text that should be corrected.
  2. "The cationic polymeric nanoparticles were designed to fulfill the two main requirements in oral delivery. One is to ensure an enhanced tolerance to acidic pH present in stomach, while the other one is to achieve an efficient phagosomal escape of bacteria from immune cells." Do these two events result from the cationic charge? 
  3. Please summarize section 2.
  4. Please add more samples to section 4.   

Author Response

The comments of this referee are deeply acknowledged since they help to improve the quality of the manuscript. In accordance with his/her main suggestions the following points have been addressed (paragraphs and pages are referred to the new manuscript):

1) There are some grammatical mistakes in the text that should be corrected.

According with the referee suggestion, the manuscript has been revised and some typos and mistakes have been corrected.

2) "The cationic polymeric nanoparticles were designed to fulfill the two main requirements in oral delivery. One is to ensure an enhanced tolerance to acidic pH present in stomach, while the other one is to achieve an efficient phagosomal escape of bacteria from immune cells." Do these two events result from the cationic charge?

These two events are, in fact, a result of both cationic and polymeric features of the coating of bacteria. On one hand, the authors of this work describe that the coating of live bacteria with cationic polymeric nanoparticles provides bacteria with a protective nanoparticle coating layer that significantly enhance the acid tolerance of bacteria in stomach and intestines. On the other hand, the (poly)cationic nature of this nanoparticles coating is ultimately responsible for facilitating bacteria to effectively escape phagosomes.

To clarify this point, the paragraph has been modified as follows (page 11): “The polymeric coating of bacteria provides an enhanced tolerance to acidic pH present in stomach, while the polycationic nature of this coating facilitates an efficient phagosomal escape of bacteria from immune cells.”

3) Please summarize section 2.

The use of nanoparticles for immunotherapy is a complex field and for this reason, we have included in this section the basic principles of the antitumoral immune response with a summary of the current state of the art of the use of nanoparticles for immunotherapy. To perform a simplification in this section is complicate because some important information could be missed. In any case, according with the reviewer suggestions the section 2 has been slightly simplified removing a few paragraphs.

4) Please add more samples to section 4. 

According with the referee suggestion, we have included more examples of the use of bacteria as nanoparticle carriers (section 4.4, pages 13-14):

Another strategy for tumor regression is the use of a combination therapies combining PDT with immunostimulation. In the work reported by Lanlan et al.[120], the photo-synthetic bacteria Synechococcus 7942 were employed for the delivery of a photosensi-tizer to the tumoral mass for in situ oxygen generation to achieve photosynthe-sis-boosted PDT. For this, human serum albumin nanoparticles were loaded with the photosensitizer indocyanine green (HSA/ICG) and they were covalently attached onto the surface of bacteria by amide bonds to form the nanosystem S/HSA/ICG. Once injected intravenously into tumor-bearing mice, S/HSA/ICG effectively accumulated in the tumor and gener-ated oxygen when 660 nm laser was irradiated. The continuous production of oxygen remarkably ameliorated tumor hypoxia and promoted reactive oxygen species production, thereby inducing tumor cell death. Moreover, this nanocarrier can also effectively revert the tumor immunosuppressive microenvironment and increased antitumor immune responses by generating IL-10, increasing IFN-γ production, and inhibiting production of the immunosuppressive mediator TGF-β. S/HSA/ICG showed an optimal effect in preventing tumor recurrence and inhibiting metastasis in mouse model of metastatic triple-negative breast cancer. This work represents an interesting approach for enhancing immunogenic PDT with photosynthetic bacteria for tumor remission.

Combination of immunotherapy with additional functions as PDT or photothermal therapy (PTT) allow generation of synergistic effects that enhance the therapeutic outcomes. Similarly to the previous work, Chen and coworkers designed a nanosystem [126] for a triple therapy based on biocompatible polydopamine nanoparticles coated on the surface of anaerobe Salmonella VNP20009. The coated nanosystem was intravenously injected followed by irradiation with a near-infrared laser at the tumor zone and combined with a local administration of a phospholipid-based gel loaded with the anti-PD-1 peptide AUNP-12. This gel releases in a sustainably way the AUNP-12 peptide for 42 days, thus maintaining the tumor microenvironment as immunopermissive. The triple combination of PTT, biotherapy, and PD-1 blockade showed high efficacy in achieving strong antitumor immune responses and eliminating large tumors in 50% of animals from a mouse model of melanoma up to 80 days. These results yield new insights about the advantages of BMTT, and this innovative strategy may represent a powerful anticancer immunotherapy tool.”

Reviewer 3 Report

The authors should mention the properties and application of bacteria that are used as nanocarriers.

The aim of this review article also missing in this article.

The biological pathways of bacteria and cancer cells should be explained.

How can you expect the stability of bacterial nanocarriers for cancer immunotherapy?

The discussion section is faded and own ideas of the authors should be mentioned.

I think the authors should arrange the subtitle sequentially

Author Response

The comments of this referee are deeply acknowledged since they help to improve the quality of the manuscript. In accordance with his/her main suggestions the following points have been addressed (paragraphs and pages are referred to the new manuscript):

1) The authors should mention the properties and application of bacteria that are used as nanocarriers.

The authors want to thank the reviewer for pointing out this issue. We have described along the section 3 and enumerated from 1 to 5 the main properties of bacteria in antitumoral therapy. However, for a better comprehension, we have also included the next paragraph (page 10): “Overall, the required properties of bacteria to become appropriate nanocarriers are high motion capacity and tumor chemotaxis. The motility is provided by flagella, while chemotaxis can be provided by different characteristics present in tumors, such as presence of hypoxia or tumor-related metabolites as lactic acid excess. Thus, only flagellated and hypoxia- or nutrient-attracted microorganisms are useful as nanocarriers in cancer therapy. Attenuated strains of E. coli, Salmonella or Clostridium fulfill these essential requirements, so the works described below mainly use these bacteria as effective carriers.”

2) The aim of this review article also missing in this article.

The aim of this review is to describe the recent advances carried out in the application of bacteria as nanoparticle carriers for antitumoral immunotherapy. This is a complex field which involve different fields and for this reason we have included a section which address the basic principles of antitumoral immune response, the use of nanoparticles to trigger efficient immune responses, the use of bacteria as nanoparticle carriers and finally, the combination of bacteria and nanoparticles in immunotherapy.  In order to clarify the aim of this review, the following paragraph has been included: (Page 2) “In this work, the recent advances carried out on the development of bacteria as nanoparticle carrier for antitumoral immunotherapy will be presented to describe a powerful strategy to induce sustained and efficient immune responses against tumors.” 

3) The biological pathways of bacteria and cancer cells should be explained.

The authors want to thank the reviewer for pointing out this interesting issue. For a better explanation of this pathways, the next paragraphs have been added (pages 8-9): “The biological interactions stablished between cancer cells, bacteria, and the sur-rounding tumor microenvironment provoke complex alterations in tumor-infiltrated immune cells and in chemo/cytokines, which, in turn, promote tumor regression.[72] Tumor regression can be induced by several different mechanisms once bacteria have colonized and growth inside the tumoral tissue. Depending on the bacterial strain, the display of tumor suppression mechanisms within the tumor microenvironment is dif-ferent. One mechanism is the in-situ production of bacterial toxins of Salmonella, Lis-teria or Clostridium, that directly kills tumor cells by inducing apoptosis or autopha-gy.[86] For example, Salmonella toxins induce upregulation of the ubiquitinated protein Connexin 43 in tumor cells, promoting the formation of junctions between malignant cells and dendritic cells. These functional connections allow cross-presentation of tu-mor antigens to dendritic cells, leading to reduced expression of the immunosuppres-sive IDO enzyme in T-cells and a consequent and specific increase in CD8+ T-cell acti-vation.[87]

Concerning the host–pathogen interaction, the rapid proliferation of bacteria in tu-moral tissues and the presence of bacterial components as lipopolysaccharide (LPS) and flagellin induce significant migration of innate immune cells such as macrophag-es, neutrophils, and dendritic cells towards colonized tumors. Then, inflammasome ac-tivation leads to a strong production of interleukin-1β by macrophages and dendritic cells.[88] LPS is involved in the high secretion of TNF-α cytokine via interactions with CD14 and TLR4 receptors.[89] Salmonella flagellin is involved in inflammasome-driven secretion of interleukin-1β and interleukin-18, which serve as activators of IFN-γ, a cytokine that induce the production of cytotoxic T cells and natural killer (NK) cells.[90]”

4) How can you expect the stability of bacterial nanocarriers for cancer immunotherapy?

We agree with the reviewer that the stability of bacterial nanocarriers is an important aspect that must be taken into consideration for cancer immunotherapy. However, we find extremely difficult to assess an overall or a general statement about the stability of this type of nanosystems. The stability depends on the inherent features of each bacterial nanocarrier: bacteria type, nanoparticle nature, surface charge, existence of PEG or other targeting moieties, etc. Indeed, there are many biological factors that may affect the stability of the nanosystem, as macrophages capture or the type of bond between the bacterium and the nanoparticle (i.e. the stability will be different depending if the stablished bond is covalent, acid-labile, or simply ionic interactions). Some of these concerns have been addressed in each case in the description of the different examples discussed along the manuscript.

5) The discussion section is faded and own ideas of the authors should be mentioned.

The authors respectfully disagree with the reviewer about this point. For example, along section 3, we have described the main characteristics that makes bacteria an appropriate tool for immunotherapy against cancer. The own ideas about the state of the art of the field have been discussed along the manuscript, both regarding the use of nanoparticles in antitumoral immunotherapy and the advantages and potential problems of the use of bacteria as nanoparticle carriers.

6) I think the authors should arrange the subtitle sequentially.

We have designed the manuscript to follow a thread from the use of nanomaterials as drug delivery systems of immunotherapeutic agents to the application of bacterial nanocarriers of nanomaterials for enhanced anti-tumor immunotherapy. According with the reviewer suggestion, in order to make bacterial nanocarriers section 4 clearer and easy to follow for the readers, we have included the subsections 4.1, 4.2, 4.3 and 4.4 to organize the information and to facilitate its easy location along the section. Additionally, the following paragraph have been added:

Page 10: “Immunotherapy-based strategies to fight cancer are gaining more and more importance. In recent years, several works using bacterial nanocarriers for immunomodulation have been presented. Herein we describe some of these works based on the different approach employed.

Round 2

Reviewer 3 Report

Thank you very much for your prompt responses and sound updated manuscript. I want to appreciate the authors' research responsibility.